# Novel Method for Electroless Etching of 6H–SiC

**DOI:** 10.3390/nano10030538

**Published:** 2020-03-17

**Authors:** Gyula Károlyházy, Dávid Beke, Dóra Zalka, Sándor Lenk, Olga Krafcsik, Katalin Kamarás, Ádám Gali

**Affiliations:** 1Institute for Solid State Physics and Optics, Wigner Research Centre for Physics, Konkoly-Thege Miklós út 29-33, H-1121 Budapest, Hungary; beke.david@wigner.hu (D.B.); zalka.dora@wigner.hu (D.Z.); kamaras.katalin@wigner.hu (K.K.); 2Department of Atomic Physics, Budapest University of Technology and Economics, Budafoki út 8, H-1111 Budapest, Hungary; lenk@eik.bme.hu (S.L.); krafcsik@eik.bme.hu (O.K.)

**Keywords:** chemical etching, silicon carbide, nanoparticles, X-ray photoemission spectroscopy, photoluminescence spectroscopy

## Abstract

In this article, we report an electroless method to fabricate porous hexagonal silicon carbide and hexagonal silicon carbide nanoparticles (NPs) as small as 1 nm using wet chemical stain etching. We observe quantum confinement effect for ultrasmall hexagonal SiC NPs in contrast to the cubic SiC NPs. We attribute this difference to the various surface terminations of the two polytypes of SiC NPs.

## 1. Introduction

Silicon carbide has received attention in the last decades due to its unique mechanical and chemical properties that make it a promising candidate for various applications including solar panels [1,2], quantum technology [3,4,5,6,7,8,9] and bioimaging [10,11,12,13,14]. Silicon carbide is a polytypic material in which polytypes share the same basal plane unit cell but have different stacking of Si–C bilayers along the c axis. These polytypes contain the same Si–C bonds, however, the variety of the band gaps (2.4 eV in cubic 3C polytype versus 3.0 eV in hexagonal 6H polytype and 3.2 eV in hexagonal 4H polytype) may lead to different electrochemical processes by applying the same etchant. Pristine 3C–SiC can be etched by a mixture of nitric acid and hydrofluoric acid [15]. This method, however, is only applicable to the 3C polytype [16]. So far, porous hexagonal SiC has been prepared by electrochemistry [17] or metal assisted etching [18] or a complex method based on those two [19], or etching in alkaline solution [20,21,22], which requires n type [23,24] or p type [25] silicon carbide as a bulk material. This approach poses a restriction on the purity of the material as either p type or n type SiC can be achieved by doping. Thus, there is an urgent need for finding a method which allows the preparation of porous SiC by electroless etching. In our previous study [26], we proposed a new method that could give us better insight and understanding of physico-chemical processes behind the stain etching of semiconductors, which is a common method to obtain nanostructures. The “no photon exciton generation chemistry” (NPEGEC) suggests that the position of the band edge with respect to the potential of the reactants is crucial. This method paves the way for finding suitable reactants for other polytypes such as 6H–SiC.

In this article, we demonstrate that a mixture of sodium dithionate and hydrofluoric acid can efficiently stain etch hexagonal 6H–SiC. During the reaction, porous 6H–SiC is formed, that is subsequently transformed to nanoparticles. The colloid samples exhibit sharp size distribution and ultraviolet luminescence caused by quantum confinement.

## 2. Materials and Methods 

Several different sets of parameters were used for varying the porosity; finally, we chose the set with the highest yield in nanoparticles. In a teflon digestion chamber, 2.5 g of 6H-SiC (Sigma Aldrich, St. Louis, MO, USA, 200–450 mesh particle size), 2.5 g of Na_2_S_2_O_6_·2H_2_O (Reanal Kft, Budapest, Hungary) and 30 mL of hydrofluoric acid (VWR International Kft, Debrecen, Hungary, Analar Normapur, 48%) were blended. The mixture was heated up to 150 °C and was kept at this temperature for 2 h. The remaining excess of hydrofluoric acid was removed by repeatedly elutriating, decanting the liquid phase, then diluting with distilled water. Sulphuric compounds were eliminated by excessive rinse of the porous SiC with large amounts of distilled water and toluene (VWR International Kft, Debrecen, Hungary, Hipersolv Chromanorm). The product was added to approximately 100 mL water and was sonicated for 2 h. During the process, SiC NPs were obtained by mechanically destroying the porous layer. The NPs were separated from the bulk material by centrifuging at 4500 rpm for 30 min (Thermo Fisher Scientific, Waltham, MA, USA, Heraeus Megafuge 8). The supernatant was once again decanted and pressed through a Pall Acrodisc 32 mm syringe filter with 0.1 µm Supor Membrane (Sigma Aldrich, St. Louis, MO, USA) to eliminate any remaining bulk material. The obtained colloid dispersion of SiC NPs was separated by size using a centrifugal filter. Photoluminescence (PL) measurements were carried out on these samples using a HORIBA Jobin-Yvon Nanolog FL3-2iHR fluorometer (Horiba Ltd., Kyoto, Japan) equipped with a 450W Xenon lamp, iHR-3210 grating system, and a R928P photomultiplier tube for the measurement in the ultraviolet (UV) and visible range. Using either aluminum or silicon wafer, we evaporated droplets of samples for scanning electron microscopy (SEM) (TESCAN, Brno, Chech Republic, TESCAN MIRA3)and atomic force microscopy (AFM) (Bruker France., Palaiseau, France, Dimension Icon) measurements. Droplets on silicon wafer were also used for infrared (IR) measurements on a Bruker Tensor 27 instrument (Bruker Corp, Billerica, MA, USA). The x-ray photoelectron spectroscopy (XPS) measurements were carried out using a twin anode X-ray source (Thermo Fisher Scientific, Waltham, MA, USA, XR4) and a hemispherical energy analyzer with a nine-channel multi channeltron detector (SPECSGROUP, Berlin, Germany, Phoibos 150 MCD). The base pressure of the analysis chamber was around 2 × 10^−9^ mbar. Samples were analyzed using a Mg Kα (1253.6 eV) anode, without monochromatization. For analyzing bulk samples we used Focused Ion Beam-Scanning Electron Microscope (FIB-SEM) (Thermo Fisher Scientific, Waltham, MA, USA, FEI Quanta D3), and energy-dispersive spectroscopy (EDX) (AMETEK Inc., Berwyn, PA, USA, Edax Ametek Element detector controlled with APEX software package).

## 3. Results

### 3.1. Chemical Etching and Morphology of the Bulk 6H-SiC

Appearance and morphology of the untreated, etched or sonicated bulk samples were investigated by SEM. It can be clearly seen from the image that the grains of the untreated sample have intact, whole edges with smooth flats and only few conchoidal fracture can be observed (Figure 1a). In the case of the dithionate-hydrofluoric acid treated sample, an increased surface porosity and the erosion of the edges can be observed (Figure 1b). It can be suggested that the damage of the edges starts along the crystal dislocations. 

Repeated etching causes more damaged and porous flats (Figure 1c,d). Dual beam FIB-SEM systems were used to cut through the sample and make the cross section visible for further investigation of the porosity. 

The cross-section image of the untreated sample shows an undamaged plane and no internal porosity can be observed (Figure 2a). On the other hand, in the case of the etched sample, increased porosity has been verified (Figure 2b). This result confirms the efficiency of the acid treatment and provides further support for the NPEGEC. Previously our research group managed to stain etch cubic SiC and used that porous material to fabricate nanoparticles using reactants that were selective for 3C–SiC [26]. In order to compare the 3C and the 6H polytypes, two samples were etched by their dedicated reactants and were investigated by SEM/FIB. From the SEM images, it can be concluded that the untreated 3C–SiC surfaces have higher porosity than untreated 6H–SiC, and small pores (100–400 nm) can be observed (Figure 2c,d) in the cross section. Even though the pore size and grain size are different for the two polytypes, the cross-section images of the etched 3C–SiC are similar to the etched 6H–SiC sample cross section. Based on this similarity, we expected the etched 6H–SiC sample to be appropriate for fabricating 6H nanoparticles. See Appendix A for a mathematical estimation on the porosity based on high precision weight measurement.

### 3.2. Characterizing the 6H Nanoparticles

SEM measurements were carried out for the detection of the NPs and for the determination of their size. Results indicated the presence of significant amounts of contaminations. SEM-EDX measurements confirmed that the solution contained sulphuric compounds as byproducts. We assume that this is due to the increased porosity of the bulk material, which means that our cleaning methods need further improvement. Nevertheless, we successfully managed to separate the contaminations from the product in liquid phase. Using centrifugation, we obtained two samples. The permeate was enriched in the contaminations whereas the retentate was free of dithionate and fluoric compounds. This also proves that the retentate did not contain carbon fluorooxide nanoparticles [27].

In both cases, the size distribution was measured by AFM (see Figure 3a–d) and SEM (not shown).

The cutoff of the membrane is at 4–5 nm. The fractions will be marked and identified by their average size in nanometers. In the permeate, one can observe two fractions: a smaller one at ca. 3 nm and a larger one at 12–15 nm.

In good agreement with the expectations, the retentate had larger particles: the small fraction was at ca. 5 nm and the larger fraction was at 25–30 nm. We note that for the measurements we evaporate droplets of the samples. We suggest that in both cases larger fractions are observed as a combined result of aggregation and possible stacking of unique particles during the drying process, which could appear as a single particle and thus, would create seemingly larger particles.

To identify the composition of the nanoparticles, we applied XPS. Peak fitting was carried out using CasaXPS software. Approximately 2 mL of the retentate was dried on a small niobium wafer. First, we identified the adventitious carbon in the spectrum and shifted the spectrum by 3.9 eV so the peak position matched the literature data. Presumably the electric contact between the wafer and the semiconductor nanoparticles is inadequate. This causes a different charge for the SiC containing dried layer. This is confirmed e.g., by the presence of N 1s peaks with a difference of 1.4 eV. Assuming that these twin peaks emerge from the charged sample and the insulating island of the nanoparticles simultaneously, an additional 1.4 eV shift was applied for the peaks associated with SiC. The spectrum confirmed the presence of carbon and silicon in the sample.

The silicon (Si 2p) peak could be fit by a single gaussian at 100.6 eV, which is in perfect correlation with the Si–C bonding energy [28,29], see Figure 4a. Multiple carbon peaks (C 1s) were observed, (Figure 4b) some of which can be associated with C–Si bonding. In the case of the carbon spectrum, it was not obvious which fit components should be shifted by 3.9 eV and which ones should be shifted by 5.3 eV in total.

We did a total permutation and managed to associate the fit components to specified bonds as listed in Table 1.

This interpretation is not strongly conclusive because the reported carbon XPS peaks associated with C–Si bonding scatter between 281.3–283.4 eV [30,31]. Nevertheless, the presence of Si–C bonds is evident from the XPS analysis and it confirms the composition of the nanoparticles.

Additionally, we detected photoluminescence from the colloid samples. We used liquid phase samples for the measurements so we can assume that aggregation does not occur in these cases. We note that sonicating the as-received bulk SiC or the absence of any of the reactants resulted in liquid supernatant samples that showed no luminescence at all. In those blinded experiments, SEM could not confirm the presence of nanoparticles either. PL spectra of the etched and sonicated 6H–SiC sample is shown in Figure 5a. A strong peak with a maximum between 2.7 eV and 3.1 eV can be observed. The broad peak extends above 3.0 eV, which is the band gap energy of the 6H polytype. This means that the sample shows above bandgap luminescence [32]. The position of the peak showed excitation wavelength dependence.

The PL spectra of the permeate and the retentate are shown in Figure 5b,c. The spectra of the retentate are quite similar to those of the unseparated sample. On the other hand, the luminescence of the permeate peaks is at 3.0 eV.

Our measured spectra showed that the emission wavelength with the maximum intensity depended on the excitation wavelength. When comparing to 3C–SiC NPs, one can expect the arising luminescence to be surface dominated in particles smaller than 4 nm of diameter [33]. However, we could observe this phenomenon both in the retentate and in the permeate, and the permeate did contain particles less than 4 nm of diameter. It is safe to assume that luminescence arises from unique particles and the difference between the luminescence peaks is due to quantum confinement.

This suggests that the surface of the 6H nanoparticles differs from the surface of the 3C nanoparticles, thus the transition phase between bulk luminescence and surface dominated luminescence may occur at another size.

In order to examine this phenomenon, we performed IR measurements on our samples. While it is complicated to identify the functional groups of the 6H NPs, it is obvious that the spectra significantly differ from those of the 3C NPs (See Appendix A). For further testing, we applied 180 °C heat treatment to the sample and measured the spectral change over time. For 3C–SiC NPs, we successfully observed the desorption of water and the formation of anhydride groups at this elevated temperature [29]. For 6H–SiC NPs, the hydroxyl group band at 3250 cm^−1^ became stronger over time. The band associated with the C=C group at 1610 cm^−1^ and the C–H bonds at 2800–3200 cm^−1^ decreased over time in strong correlation with the emerging hydroxyl band (See Appendix A). Having several isosbestic points in the spectra practically rules out the possibility of decomposition of the sample. These results further support our assumption of the fundamentally different surface termination of cubic and hexagonal SiC NPs and provides an additional argument for the transition phase between bulk and surface dominated luminescence in 3C–SiC NPs versus only bulk luminescence in 6H–SiC NPs.

## 4. Conclusions

We successfully created porous SiC from bulk 6H–SiC using electroless wet stain etching avoiding the need for doped bulk material as by NPEGEC method. The increased porosity was demonstrated with SEM/FIB. Additionally we obtained nanoparticles by sonicating the porous layer and characterized their size by SEM, TEM and AFM. We observed broad luminescence arising from 6H–SiC NPs, which showed quantum confinement in a size range where the luminescence of 3C–SiC NPs is already surface dominated. We find that fewer hydroxyl groups in hexagonal SiC NPs compared to cubic SiC NPs may be responsible for this behavior.

## Figures and Tables

**Figure 1 nanomaterials-10-00538-f001:**
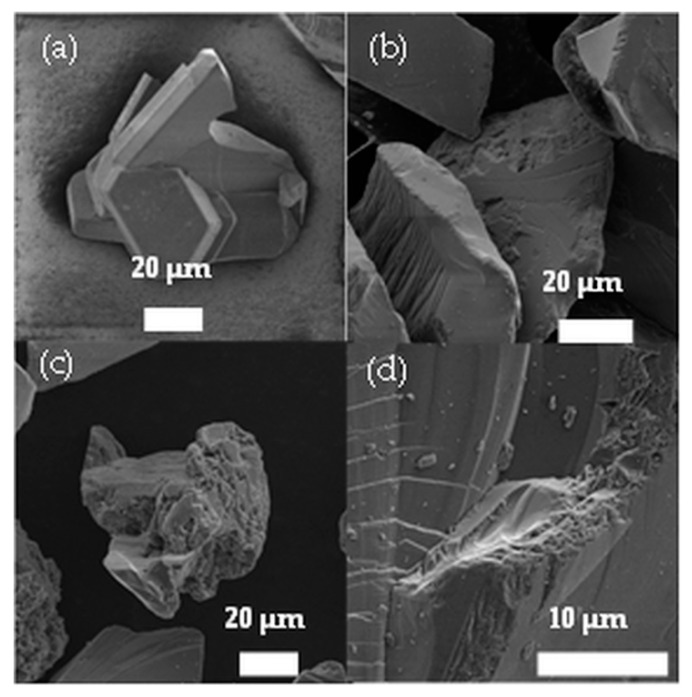
(**a**) Untreated 6H–SiC sample, (**b**) etched, (**c**) etched and sonicated, (**d**) double-etched sample.

**Figure 2 nanomaterials-10-00538-f002:**
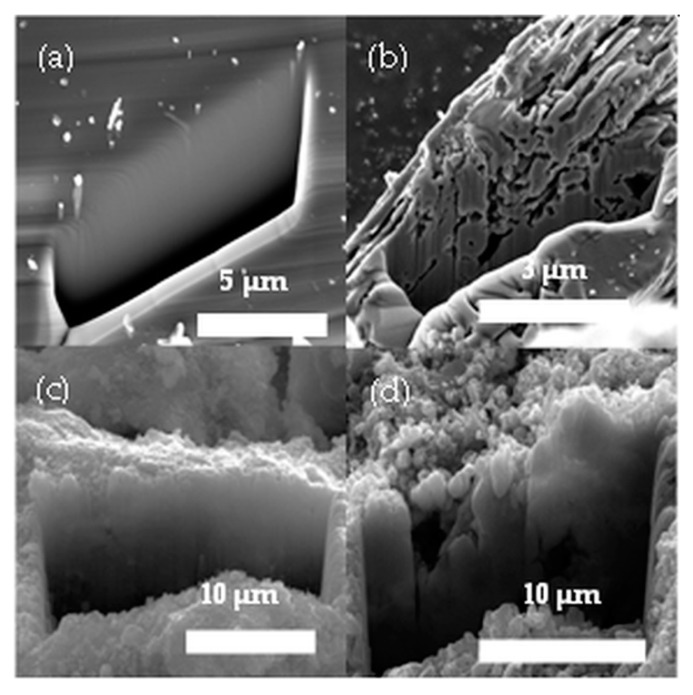
Cross-section images of (**a**) untreated 6H–SiC, (**b**) etched 6H–SiC, (**c**) untreated 3C–SiC, (**d**) etched 3C–SiC.

**Figure 3 nanomaterials-10-00538-f003:**
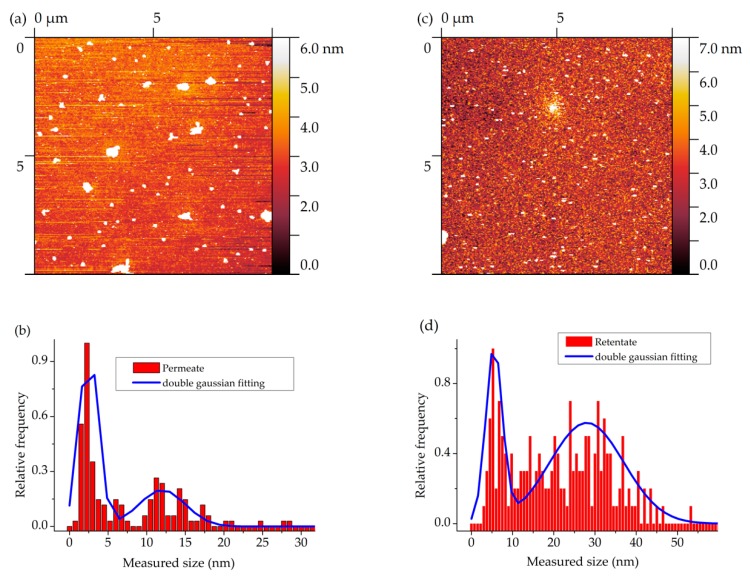
(**a**) AFM picture and (**b**) size distribution of the permeate, and (**c**) AFM picture and (**d**) size distribution of the retentate.

**Figure 4 nanomaterials-10-00538-f004:**
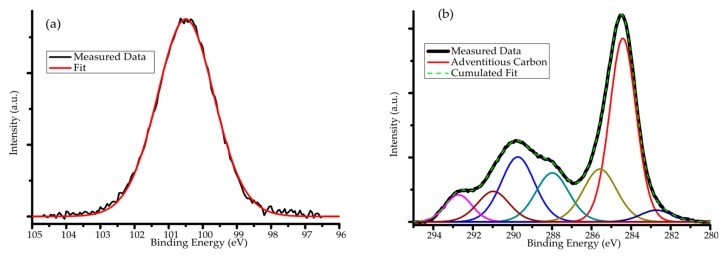
XPS spectrum and peakfit of (**a**) the Si 2p peak and (**b**) the C 1s peak in the sample.

**Figure 5 nanomaterials-10-00538-f005:**
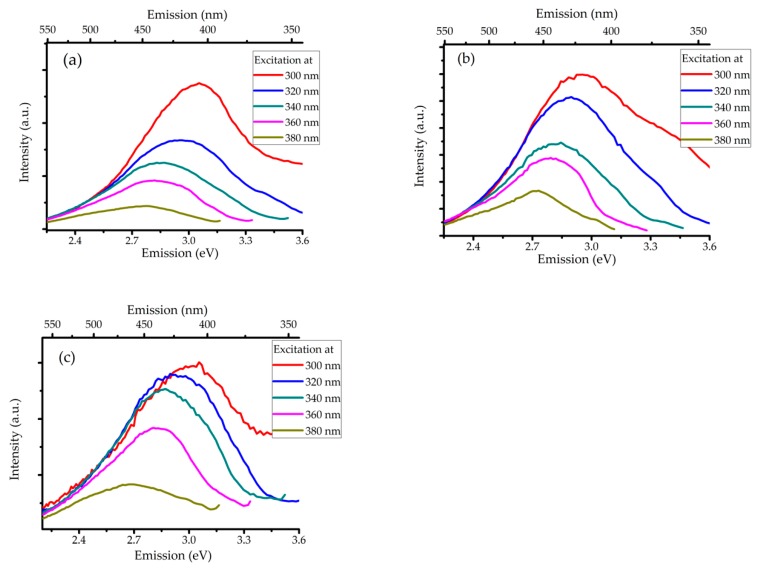
PL spectra of (**a**) the unseparated sample, (**b**) the permeate, and (**c**) the retentate.

**Table 1 nanomaterials-10-00538-t001:** Fit components and their associated bonds of the carbon spectrum.

Binding Energy	Chemical Bond or Group
284.5 eV	Adventitious carbon
289.8 eV	C=O group, C=S group
288.1 eV	C=O group
292.8 eV	C–F group
285.7 eV	C–C, C–H bond
282.8 eV	Si–C bond (shifted by 5.3 eV in total)
291.1 eV	CO_3_^2−^ ion

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
