# Peer review of "Novel Method for Electroless Etching of 6H–SiC"

_nanomaterials, 2020, doi:10.3390/nano10030538_

Round 1
Reviewer 1 Report
Abstract; 'terminators' > perhaps call it 'terminations'
Page 1, line 27; consider adding the band gap of 6H-SiC, as this value is later referred to (page 5, line 219)
Page 1, line 31; "or or"
Page 1, line 39 + Page 2, line 85; is the word "theory" justified for explanation of results ? It feels a bit exaggerated.
Page 2,line 53; Can the authors add a brief comment on the reason that toluene is used ?
Discussion of results Figure 3; can the authors comment on the number of particles detected and 'background subtraction' ? The histograms show extremely narrow 'bins' of the data, whereas the number of particles visible in the AFM images seems rather limited. How were the particles characterized, i.e. was the maximum height taken as "the" particle size ?
Page 4, line 178; What do the authors mean with 'limited resolution' ? Is it possible to quantify this statement ? The histogram in Figure 3 suggests a very high resolution. The comment on aggregation seems correct, however, the limited resolution seems odd in this context.
Page 4, line 180; what is the reason that a niobium wafer was used ? I suppose this is a Nb COATED Si wafer ?
Page 5, line 215; 'received' instead of 'recieved'
Page 5, line 215-217; can the authors explain why NO luminescence was observed, i.e. not even band-gap luminescence ?
Page 5, line 218-220; "A strong peak with maximum between 2.7 eV and 3.1 eV can be observed. This means that the sample shows above bandgap luminescence [32]." > The fact that a maximum is observed at these 2 values does NOT mean that it is above band-gap luminescence. The fact that the peak is broad and extends towards the short-wavelength / high-energy side with values above the band gap, however, does.
Reviewer 2 Report
Looking for new SiC etching methods that involve electroless techniques is of interest for the scientific community.
The approach proposed in the paper is interesting but some informations are missing and some conclusions are hazardous.
In détails :
- the characteristics of the initial SiC material must be clearly deepened (cristalline structure, doping level) as the etching performance will depend on such parameters
- the obtained porosity must be investigated more than using only SEM pictures. Porosity estimation (what is "high porosity"?) and pore dimensions must be investigated as well as etching rate.
- the impurities mentionned in 3.2 must be clearly identified (particles, elements ?)
- Is there any tests that have been performed on SiC wafers?
Round 2
Reviewer 1 Report
The authors have answered all questions raised and improved the manuscript in such a way that it can be accepted in its current form.
Reviewer 2 Report
The article can be published in its present form even if some lack of information on the initial material clearly lower the impact of the study.
This manuscript is a resubmission of an earlier submission. The following is a list of the peer review reports and author responses from that submission.
Round 1
Reviewer 1 Report
Although the concept of electroless etching of chemically resistant semiconductors is of high interest, the novelty of the etching process is unfortunately not highlighted. The focus of this work is mainly on nanoparticle characterization rather than the method itself. It is recommended to include experiments on bulk wafers, combined with electrochemical measurements, to further support the electroless nature of the method.
State of the art electrochemistry of SiC could be improved. Reference 20 is not the reference referred too.
Line 74/75: If the mechanism was chemical, defect selectivity would be expected. However, in the case of an electroless mechanism such as NPEGC, etching at defect sites is less likely, as e-h recombination rates are enhanced at dislocations. This would imply that in the vicinity of defects, the etch rate is lowered, while the etching of crystalline material further away from the defect is faster, which depends on the diffusion length of the injected charge carriers. These principles are well known for defect revealing in GaN by photoetching.
XPS Si should be further worked out:
- It would be recommended to reverse the binding energy scale on the x-axis (from high to low energy).
- Si oxides need to be included in the fitting scheme, as indicated by the broad peak extending to higher binding energies. There is Si(1+), Si (2+) and possibly Si(3+).